# Drowning prevention challenges and opportunities: An exploratory study of perspectives of delegates from ASEAN nations

Amy E. Peden[1,2]*, Justin Scarr[2], Trung Doan Minh[3], Rizan Latif[4], Dao Le Thi Anh[5,6], Tan Lii Chong[7], Delphine Fong[8], Geh Cheow Mei[9,10], Adisak Suvanprakorn[11], Sirirat Suwanrit[12], Geh Cheow Chien[9], Jose Arne A. Navarra[13], Agus Dharma[14], Thuan Tek Geh[9], Bouathep Phoumin[15], Belinda Lawton[2]

1 School of Population Health, UNSW Sydney, Kensington, NSW, Australia, 2 Royal Life Saving Society– Australia, Sydney, Australia, 3 Swim Vietnam, Hoi An City, Quang Nam Province, Vietnam, 4 Beach Bunch, Jerudong, Brunei, 5 Hue University of Sciences, Thành phố Huế, Tha Thiên Huế, Vietnam, 6 Hue Help, Thuận Thành Thành phố Huế, Tha Thiên Huế, Vietnam, 7 Singapore Life Saving Society, Singapore, Singapore, 8 Sport Singapore, Singapore, Singapore, 9 Life Saving Society Malaysia, Penang Life Saving Society Malaysia, Tanjung Bungah, Penang, Malaysia, 10 Universiti Sains Malaysia (USM), Gelugor, Penang, Malaysia, 11 Thai Life Saving Society, Bangkok, Thailand, 12 Division of Injury Prevention, Department of Disease Control, Ministry of Public Health, Nonthaburi, Thailand, 13 Philippine Life Saving TVET Academy Corporation, Bacolod City, Philippines, 14 Balawista Indonesia (Indonesian Lifeguard Association) and Udayana University, Bali, Indonesia, 15 Faculty of Medical Technologies, University of Health Sciences, Sisattanak District Vientiane, Lao PDR

* apeden@rlssa.org.au, amy.peden@my.jcu.edu.au

**Data Availability Statement:** Data are available upon reasonable request to author AEP or Royal Life Saving Society – Australia via email: info@rlssa.org.au.

## Abstract

The South East Asian region has the world's second highest fatal drowning burden. This study reports analysis of survey data from representatives from nations within the Association of South East Asian Nations regarding current efforts, challenges and future opportunities for drowning prevention. Twenty-two responses were received from respondents from all ASEAN nations excepting Cambodia and Myanmar. Drowning prevention initiatives varied across ASEAN nations, with most efforts focused on public education and raising awareness, including the provision of drowning data to the media. The lack of comprehensive, national level data collection was identified as a challenge, necessitating strengthened data collection capacity. Governmental involvement spanned one to six different ministries, highlighting the multi-sectoral nature of drowning prevention. However, a lead ministry could be identified in only two countries. Despite the challenges identified, there remain many opportunities to strengthen drowning prevention across ASEAN nations, addressing a significant regional public health threat.

## Introduction

Drowning is a significant public health issue in the South East Asian region. In 2019, 70,034 deaths were estimated to have occurred due to drowning, the second highest number of drowning deaths across all World Health Organization (WHO) regions [1]. This estimate is

**Funding:** This research received no specific funding. Funding for the workshop was received from the Australian-ASEAN Council and further supported by Royal Life Saving Society – Australia.

**Competing interests:** The authors have declared no competing interests exist.

likely to significantly underreport the drowning burden in the region, due to the exclusion of transport-related and disaster-related drowning [2], both of which are significant contributors to mortality in the region.

Within the known burden of drowning in South East Asia, children under the age of 15 years are disproportionately represented accounting for 33% of all drowning deaths [1]. Drowning in the region has a strong socio-economic gradient, disproportionately impacting the poorest members of the community, as well as minority populations [3]. The reliance on water transportation across the region, as well as the frequency and context of exposure to hazards like extreme weather events, including cyclones, tsunamis and floods, significantly increase drowning risk [1].

There are several promising initiatives that aim to reduce drowning burden in the region, including: swimming and water safety skills training in Bangladesh [4], India, Nepal and Sri Lanka; anchals (community daycare) in Bangladesh and India; and improved data collection and awareness raising campaigns in Thailand [5]. However, the implementation and scale-up of interventions known to be effective in reducing drowning are urgently required across the region to address this preventable cause of mortality.

Within the South East Asian region more broadly, there exists the Association of South-East Asian Nations (ASEAN), a political and economic union of 10 member states: Indonesia, Malaysia, Philippines, Singapore, Thailand, Brunei Darussalam, Viet Nam, Lao Peoples Democratic Republic (PDR), Myanmar, and Cambodia [6]. Drowning is the leading cause (after infancy) of child death and adolescent in many ASEAN nations. While some countries in the region have been trialling context-responsive drowning prevention interventions, many of the interventions are localised, and not shared across national borders despite shared risk factors.

ASEAN member states develop the ASEAN community via regional cooperation and wider dialogue involving major powers [7]. Australia as a nation, has cooperated, and continue to cooperate on many issues of mutual concern [8]. As a leader in drowning prevention, Australia often collaborates with other countries across the region, providing opportunities for skill development and knowledge sharing in the areas of drowning prevention and water safety, including via regional workshops and global conferences [9, 10]. This role exemplifies calls for collaboration and strengthening of member state responses to drowning prevention in the United Nations General Assembly Resolution on Global Drowning Prevention [11].

The Resolution also calls for a better understanding of the barriers to drowning prevention in member states, and for provision of the assistance member states needs to overcome these barriers [11]. In response to this call to action, Royal Life Saving Society–Australia facilitated a regional drowning prevention workshop for representatives from ASEAN nations. Prior to the workshop, and at the conclusion of the event, participants from ASEAN nations were surveyed. The aim of this study therefore, was to identify the barriers to implementing the United Nations General Assembly resolution on global drowning prevention, as well as opportunities for cross-seeding drowning prevention ideas and opportunities in ASEAN nations.

## Methods

This study reports the findings of surveys completed before and after a drowning prevention workshop and forum for representatives from ASEAN nations, held on May 29–31, 2023, in Penang Malaysia with the support of the Fire and Rescue Department of Malaysia be(*Jabatan Bomba dan Penyelamat Malaysia)*, known as BOMBA (under the purview of the Ministry of Local Government, Development and the Water Safety Activity Council of Malaysia) and the Life Saving Society Malaysia Penang [12].

## Background to workshop and forum

The purpose of bringing together government and non-government representatives from the region was to facilitate information and knowledge sharing, building networks and to promote a better understanding of interventions which are effective at preventing drowning mortality and morbidity.

Resourcing is known to be a significant barrier preventing this group attending conferences and workshops. Funding was secured from the Australia-ASEAN Council to Royal Life Saving Society–Australia to assist in covering costs including airfares and accommodation. The individuals' organisation was responsible for covering salary and any per diems. Royal Life Saving Society–Australia, Life Saving Society Malaysia Penang and BOMBA covered the remaining costs.

## Content of workshop and forum

Over three days, participants gathered for a workshop and forum. The first two days were structured to maximise opportunities for information sharing between participants, and learning based on the needs that were identified by participants in the first survey.

The World Health Organization representative highlighted the known drowning burden, resources and reports, and the implications of the World Health Assembly Resolution WHA76.18 'Accelerating Action on Global Drowning Prevention' which was being debated concurrently to the workshop [13]. One person from each of the nations within ASEAN represented gave a brief situational assessment of the context for drowning prevention in their country.

Other speakers provided in-depth presentations on specific interventions and skills building activities in topics including drowning data identification, media and social media use, and stakeholder identification with exemplars of collaboration.

On the third day BOMBA convened a national drowning prevention workshop (conference) to highlight work underway in Malaysia, and further strengthen exchange across nations and among multisectoral actors from Malaysia.

## Participants

The list of invitees was compiled from existing networks of both WHO and RLSSA, and using publicly available documents including World Health Organization reports, a list of relevant government departments and non-government organisations known to be active in drowning prevention.

Within limited financial resources, the aim was to identify at least one individual from a government department and one from a non-government organisation from each of the target nations, with the goal of maximising country representation and bringing together stakeholders with relevant knowledge and policy influence.

## Survey development

Two anonymous surveys were developed, one for completion prior to the workshop and one circulated immediately after the conclusion of the workshop. Surveys were developed by a drowning prevention researcher (AEP) and an RLSSA employee tasked with workshop planning and execution (BL). Both surveys included a participant information sheet at the beginning of the survey, which also indicated consent to participate was implied by survey completion and submission.

The pre-workshop survey included 20 questions across five sections: i) Demographics (i.e., country of residence, age group, and gender); ii) Drowning Prevention Activities (i.e., activities in own country, activities in own organisation, government ministry involvement and main government responsibility for drowning prevention).; iii) Drowning Prevention Plans and Drowning Data Collection (i.e., existence of plan and entity tasked with leading it, presence format, content and use of drowning data, current challenges and future opportunities associated with drowning data collection, analysis and/or reporting; iv) Workshop Planning (i.e., identification of the topics participants wanted to hear about at the workshop and what they hoped to learn by the workshop's completion); and v) Final thoughts on drowning challenges and opportunities in their country or more broadly for ASEAN nations.

The post-workshop survey comprised 15 questions. These included repeated demographic questions from survey one, evaluation of the workshop and biggest challenges moving forward in preventing drowning in their country. Participants were then asked to indicate if there was something they planned to do as a result of what they learned at the event to further drowning prevention efforts in their county. For those who said yes, they were asked to describe what their plans are.

## Data collection

Surveys were hosted online using Qualtrics. All respondents were sent the same link to ensure the survey was anonymous. The pre-workshop survey was sent approximately four weeks prior to the event and was open for three weeks during which time, two reminder emails were sent to encourage responses. The invitation to complete the post-event survey was sent out to workshop participants via email on the day after the event concluded and was open for two weeks with one reminder email sent.

Surveys were developed in English language, however, aside from the closed ended questions, participants were encouraged to respond in their native language. Only one respondent to the pre-survey did so, and Google Translate was used to translate responses into English for analysis.

## Data analysis

Once data collection was complete, survey data were downloaded from Qualtrics in Excel and SPSS format. As drowning is an emerging area of work within the region, the socio-demographic information of respondents across both surveys were combined to preserve anonymity when displayed in the manuscript.

Analysis comprised descriptive statistics of quantitative data and content analysis of qualitative data captured via open-ended questions and associated free text responses [14]. Content analysis comprised identification of commonly occurring ideas and grouping into categories. Select quotes are used verbatim to illustrate codes, with edits made for grammatical purposes depicted in square brackets.

## Ethics approval

Ethics approval was granted by the University of New South Wales Human Research Ethics Committee (approval number HC230180). Participant written consent was obtained via the first question of the surveys.

## Results

In total, there were 30 people invited to complete the surveys (15 for the pre and post surveys respectively). A combined total of 22 responses were received, an overall response rate of 73%

**Table 1. Socio demographic information of respondents to workshop surveys (N = 22).**

|  | Number | Percent (%) |
|---|---|---|
| Total | 22 | 100.0 |
| Pre Survey | 9 | 40.9 |
| Post Survey | 13 | 59.1 |
| Gender |  |  |
| Female | 7 | 31.8 |
| Male | 15 | 68.2 |
| Age group |  |  |
| 18–34 years | 2 | 9.1 |
| 35–44 years | 0 | 0.0 |
| 45–54 years | 9 | 40.9 |
| 55+ years | 11 | 50.0 |
| Country |  |  |
| Brunei Darussalam | 2 | 9.1 |
| Indonesia | 2 | 9.1 |
| Lao PDR | 2 | 9.1 |
| Malaysia | 3 | 13.6 |
| Philippines | 2 | 9.1 |
| Singapore | 3 | 13.6 |
| Thailand | 2 | 9.1 |
| Viet Nam | 6 | 27.3 |

(60% pre survey and 87% for the post survey). Responses were received from all countries who participated in the workshop aside from Malaysia. All ASEAN nations were represented at the workshop aside from Cambodia and Myanmar. Socio-demographic information of respondents can be found in Table 1.

## Current activities

At the national level, respondents identified the most common drowning prevention activity being undertaken across ASEAN nations was awareness raising and public education activities (55.6%). One respondent described the expansion of efforts as "*The national television broadcasts an education program on drowning prevention for children. There are several organisations working on drowning prevention, they spread useful information related to drowning prevention on social media*". Other prominent activities were: strengthening multi-sectoral collaboration (33.3%); provision of swimming lessons for children (33.3%); and lifeguard training (33.3%). The next tier of activities cited by respondents included: legislative or policy-based approaches (22.2%); community rescue training (22.2%); and survival swimming instructor training (11.1%).

At an organisational level, respondents surveyed were commonly involved in the provision of swimming and water safety education for children (44.4%). As one respondent described "*Our organization has a main program called Swimming For Safety. We train schoolteachers survival swimming and water safety awareness, then the teachers will teach children. We also provide free water safety education for children aged 6–15. We partner with other organizations to have trainings on a national level*". The next most common activities respondents were involved in were: community education (33.3%); lifeguard training (11.1%); and community rescue and resuscitation (11.1%).

## Government involvement

A range of government ministries were identified as being involved in drowning prevention across the nations surveyed including health, education, tourism, disaster risk reduction, and sport. The Ministry of Health (including public health) was the most common ministry identified in all but two countries (Indonesia and Singapore). In Brunei-Darussalam, the Philippines and Viet Nam, between four and five ministries were identified as having an involvement in drowning prevention. In Lao PDR, where drowning is an emerging issue, although Ministry of Health was identified as the aspirational lead government agency, in reality rescue volunteers are the group responsible for drowning prevention efforts to date in the country (Fig 1). Of the eight countries where representatives responded to the survey, the lead government agency for drowning prevention was identified for only two countries.

## Water safety or drowning prevention plans

In total, four respondents indicated there were national drowning prevention (Thailand, the Philippines and Viet Nam) or water safety plans (Singapore). The organisations responsible

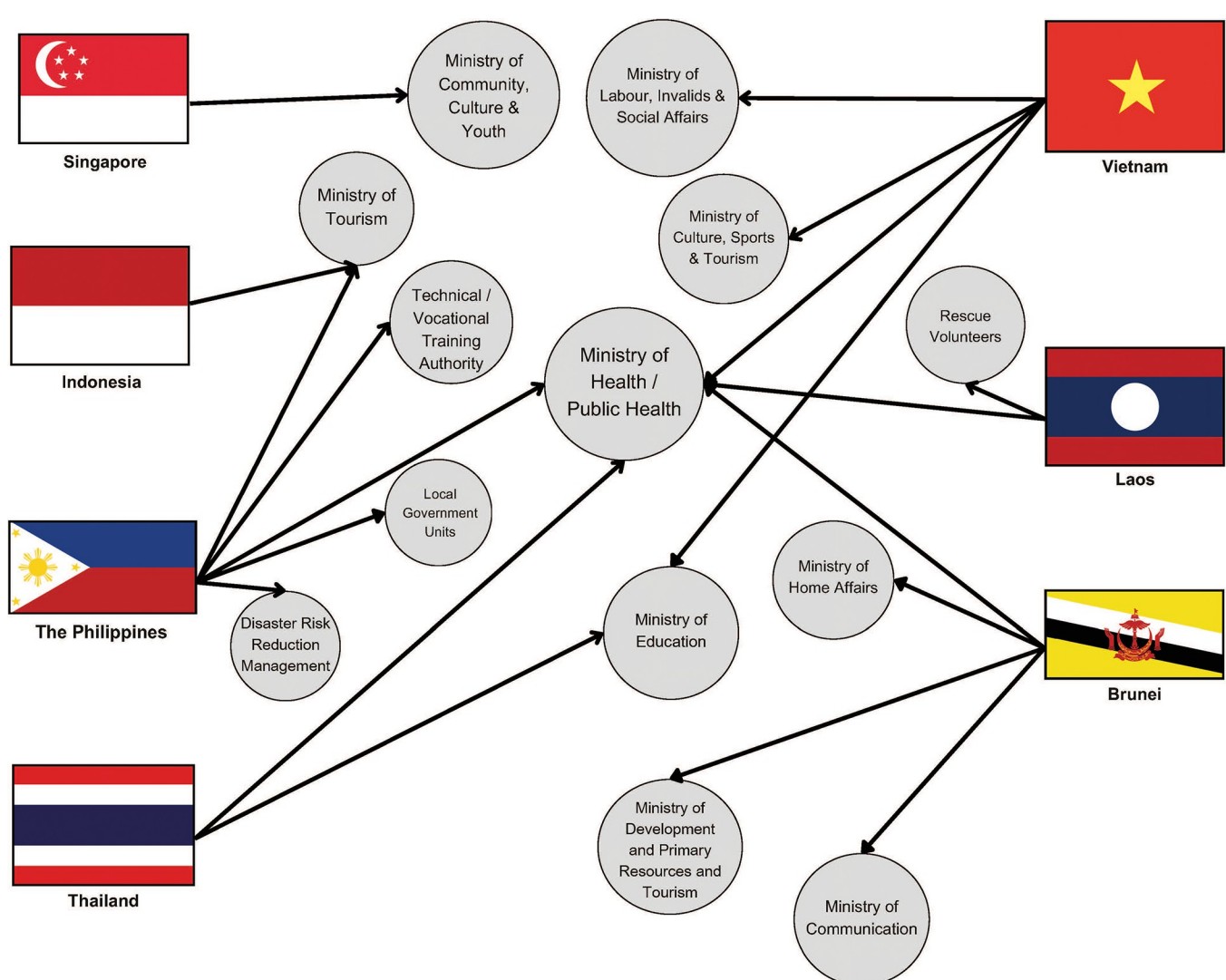

**Fig 1. Distribution of government ministries reported to be involved in drowning prevention across selected ASEAN countries.**

**Table 2. Drowning data usage purposes in selected ASEAN countries.**

| Purpose | Percentage (%) of respondents |
| --- | --- |
| Media releases or data supplied to media | 66.7% |
| In reports | 55.6% |
| In the development of drowning prevention programs or initiatives | 44.4% |
| Provided to partner organisations | 44.4% |
| In media interviews | 33.3% |
| In government submissions | 33.3% |
| In visual or animated infographics | 33.3% |
| In water safety plans or strategies | 33.3% |
| For research | 22.2% |
| In videos | 22.2% |
| In grant submissions | 22.2% |
| NGO Search and Rescue | 11.1% |

Note: No response from Malaysia received

for the development of these plans are the Ministry of Labour, Invalids and Social Affairs (MOLISA) in Viet Nam, the Philippines Drowning Prevention Council (with Philippines Life Saving Society as lead convenor) and Sport Singapore. Lao PDR and Brunei Darussalam do not have national water safety plans in place, but respondents indicated belief that there was the desire to develop a plan.

## Drowning data

Respondents indicated data on drowning was being collected in Brunei-Darussalam, Singapore, Thailand and Viet Nam. Data are not being collected in Lao PDR and respondents were unsure on data collection for drowning in the Philippines and Indonesia. Where data are being collected they are commonly being used for advocacy and public awareness opportunities via the media (66.7%), or in reports (55.6%) (Table 2).

Respondents were asked to indicate challenges around drowning data collection in their country. Several respondents identified a lack of central coordination or collection of data, which can be made more challenging in geographically large areas: "*There is no available system to collect those from the far flung areas and report at national leve*l". The availability and capacity of personnel to support the data collection task was also identified as a challenge, including the need for "*common data criteria on drowning*".

Despite these challenges, future opportunities on the topic of drowning data collection were also identified by respondents. Archipelago nations, such as Indonesia and the Philippines, recognised the value in improving data collection systems given the ever-present risk of drowning associated with inter-island travel and exposure to water. For others technology and collaboration represent opportunities for improvement: "*Taking advantage of internet technology in reporting, as well as close collaboration and sharing experiences among agencies in data collection/analysis and reporting*"; as is improving data collection from the bottom up: "*systemic data collection from the Community through to the national level*".

## Challenges in preventing drowning

After the workshop, respondents reflected on their biggest challenges in preventing drowning in their country. The most common responses were accessing sufficient funding and resourcing (30.8%), as well as government interest in, and prioritisation of the issue (30.8%).

Awareness of the issue, including in rural areas, was the next most commonly reported challenge (23.1%), followed by collaborating with other sectors, lifeguard shortages, and limited rescue equipment (7.7% respectively).

Almost all respondents intended to do something as a result of the workshop. Examples of future drowning prevention plans included "*training instructors so that they can train school teachers nationwide for survival swimming*", "*reactivation of the multisectoral drowning prevention technical working group*" and use "*World Drowning Prevention Day to have a stakeholder meeting in regards to the UNGA actions from our government*".

## Discussion

Given the significant drowning-related public health burden in the South East Asian region [1] and recent United Nations General Assembly Resolution's call to action [13], there is a need to better understand barriers and opportunities for drowning prevention. This survey of respondents from ASEAN nations aimed to examine current situation on drowning prevention, including government involvement, as well as challenges and opportunities for future action.

The multi-sectoral nature of drowning prevention has resulted in government involvement from a range of portfolios [15]. Respondents indicated ministries such as health, education, tourism and sport, among others, took an interest in drowning prevention, yet in only two countries could identify the lead ministry. As a multi-sectoral issue, political traction on the issue of drowning can be hampered if no ministry sees the issue as falling within their scope, or the issue experiences perpetual change in ministries over time [16].

A lack of political traction on the issue of drowning in some nations in ASEAN may be in part due to difficulties in quantifying the drowning burden. A lack of consolidated data on a national level, hampered by a lack of adequate resourcing and capacity to identify sources and compile the data, was identified as a key challenge by respondents. The paucity of comprehensive national data negatively impacts ability to identify emerging trends, track the impact of interventions and inform the development of National Drowning Prevention plans [15, 17]. A lack of data is also negatively affecting epidemiological research to identify risk factors and inform intervention implementation, with just 22% of countries currently using drowning data for research. This has also been highlighted as a challenge globally, with 'strengthening capacity of data systems to measure and monitor drowning burden at all levels' ranked second among fifty strategic priorities for advancing drowning prevention [18].

The exclusion of drowning deaths during disaster is a known weakness in global drowning estimates [2]. Although impossible to quantify currently, loss of life due to drowning during flood, cyclone and tsunami is likely to be significant [19]. As such, the need to better integrate the fields of drowning prevention and disaster risk reduction is evident, particularly in the context of a changing climate [20]. However, only one nation represented at the workshop and forum (the Philippines) identified disaster risk reduction within those ministries involved on the issue. This represents an area of expansion and necessary growth for other nations in the region as disasters are predicted to increase in frequency and severity [21].

### Strengths and limitations

This study, to the best of the authors' knowledge, is the first to specifically examine the interrelationship of views on the public health issue of drowning prevention among a regional cluster of ASEAN nations. Findings presented offer insights into current practice, as well as challenges to be overcome and opportunities to strengthen future drowning prevention efforts. A strength is the study surveys a mix of government and civil society actors, important for a

multisectoral issue such as drowning. The views of respondents indicate a diverse degree of engagement in drowning prevention ranging from emerging activities through to many years of action on the issue. However, this study is not without its limitations. The data presented represented the views of a small number of invited workshop participants only, with some attrition in response rates between the pre and post surveys. The small number of country-based participants made maintaining anonymity in presentation of results challenging. The survey was administered in English, which may have impacted respondents' comprehension of what was being asked of them. Surveys are also prone to inherent limitations including social desirability and recall bias and therefore results should be interpreted with caution. Consent to participate was recorded via the return of the questionnaire, which did not allow for determination of reasons for non-completion. Although reasons for non-completion aren't known, given the desire for a common strategy on drowning prevention across the region, the lack of participation from representatives of Cambodia and Myanmar in the workshop and surveys likely indicate a need to foster drowning prevention efforts in these countries. As drowning is an emerging area of work within the region, the socio-demographic information of respondents was combined to preserve anonymity. As such, multiple respondents may be reflected twice within this section which provides a description of respondents across both survey 1 and 2.

## Conclusion

Drowning burden is likely to be significant across the member nations of ASEAN, particularly given the broader scale of drowning in South East Asia. This study has identified the diversity of work already underway on the issue, as well as the need for improved data collection and the designation of a lead government ministry in many countries. Despite many challenges, the opportunities discussed represent an agenda for strengthening drowning prevention efforts and thus addressing a significant public health threat in the region.

## Acknowledgments

The authors would like to acknowledge Norman Farmer AM EMS for his work in planning and coordination of the workshop, and the Malaysia Fire and Rescue Authority for their support of, and significant attendance at, the workshop, in particular PKPjB Nordin Bin Pauzi. The authors would also like to acknowledge all presenters and contributors to the event.

## Author Contributions

**Conceptualization:** Amy E. Peden, Belinda Lawton.

**Data curation:** Amy E. Peden, Belinda Lawton.

**Formal analysis:** Amy E. Peden.

**Methodology:** Amy E. Peden.

**Project administration:** Belinda Lawton.

**Writing – original draft:** Amy E. Peden, Belinda Lawton.

**Writing – review & editing:** Amy E. Peden, Justin Scarr, Trung Doan Minh, Rizan Latif, Dao Le Thi Anh, Tan Lii Chong, Delphine Fong, Geh Cheow Mei, Adisak Suvanprakorn, Sirirat Suwanrit, Geh Cheow Chien, Jose Arne A. Navarra, Agus Dharma, Thuan Tek Geh, Bouathep Phoumin, Belinda Lawton.

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
