## [Decision Letter · Decision Letter 0]

22 Apr 2024

PONE-D-24-10409Drowning Prevention challenges and opportunities: Perspectives of delegates from ASEAN nationsPLOS ONE

Dear Dr. Peden,

Thank you for submitting your manuscript relating to a much needed field of research to PLOS ONE. After careful consideration, we feel that it has merit but does not fully meet PLOS ONE’s publication criteria as it currently stands. Therefore, we invite you to submit a revised version of the manuscript that addresses the points raised during the review process.

We look forward to receiving your revised manuscript.

Kind regards,

Jasmin C Lawes, Ph.D.

Academic Editor

PLOS ONE

Journal Requirements:

2. In this instance it seems there may be acceptable restrictions in place that prevent the public sharing of your minimal data. However, in line with our goal of ensuring long-term data availability to all interested researchers, PLOS’ Data Policy states that authors cannot be the sole named individuals responsible for ensuring data access (http://journals.plos.org/plosone/s/data-availability#loc-acceptable-data-sharing-methods).

Reviewers' comments:

Reviewer's Responses to Questions

**Comments to the Author**

1. Is the manuscript technically sound, and do the data support the conclusions?

Reviewer #1: Yes

Reviewer #2: Yes

2. Has the statistical analysis been performed appropriately and rigorously? 

Reviewer #1: N/A

Reviewer #2: Yes

3. Have the authors made all data underlying the findings in their manuscript fully available?

Reviewer #1: Yes

Reviewer #2: Yes

4. Is the manuscript presented in an intelligible fashion and written in standard English?

Reviewer #1: Yes

Reviewer #2: Yes

5. Review Comments to the Author

Reviewer #1: This study identified the barriers to implementing the United Nations General Assembly resolution on global drowning prevention, as well as opportunities for cross-seeding drowning prevention ideas and opportunities in ASEAN nations. I have some minor suggestions for further improvement.

1. The authors should articulate the design of this study, e.g., the mixed method study, co-design study or other.

2. Why the number of participants in the pre-survey was less than that in the post-survey?

3. The affiliation, professional title or research direction should be supplemented in Table 1.

4. How about the reliability of self-reported data about government involvement, water safety or drowning prevention plans?

5. The discussion and limitation sections were good.

6. The figure was not clear.

Reviewer #2: Thank you for this scholarship in a space of great need.

Title: no comment

Abstract:

Line 41 consider removing the word "grouping"

Line 50 consider removing the word "thus"

Introduction: well done

Methods:

a) As the level of scholarship in drowning prevention elevates, consenting by completion or submission of a questionnaire will become non-ideal. When these research methods are combined (in this case, the research consenting and the completion/submitting the questionnaire), it becomes unclear if the non-completion is due to non-consent, conflicts of interest or just did not complete it.

b) If I understand correctly - there were 30 people sent a pre-workshop questionnaire and that same 30 people were sent the post-workshop questionnaire. If this is not the correct interpretation, please provide clarification.

c) If the demographic information was attached to both survey submissions - due to the small number of participants from each country the data manager is likely to be able to identify the responders by country. I understand the statement in line 165 in Data Analysis about "combining to preserve anonymity" but it was unlikely that the data was anonymous to the data manager. I would agree that the data responses are anonymous to the reader of the paper. Consider making a clarifying comment about the challenges of anonymous response in a survey of this small cohort of country based participants.

Results:

a) Line 176/177 - consider sharing the more full survey response data - if I have understood the method correctly- 30 participants got sent a pre and post workshop survey = 60 surveys sent. 9 participants responded to the pre-workshop survey = 9/30 = 30% response rate. 13 participants responded to the post-workshop survey = 13/30 = 43% response rate. A total of 22 responses were received from 60 surveys sent out = 37% response rate. This may not be a weakness of the study if the method was to get a response from the each country (that did happen assuming that 2 nations did not attend).

b) The 22 responses included 2 or more participant from each of 8 of the 10 ASEAN nations.

c) A comment on the non-responses may be helpful - Did some countries delegate 2 participants to respond on behalf of the country (government and NGO) and therefore the non-responders were redundancies??

Current Activities: well done

Government Involvement:

Line 200 - consider adding heath to this list in the first paragraph.

Lone 201 - remove the comma, after ministry,

Line 203-206 - consider clarifying reworking to indicate that in Lao PDR, where drowning is an emerging issue, the Ministry of Health is identified as the aspirational lead agency, rescue volunteer agencies are active in the provision of interventions currently.

Figure 1 - I presume is the nations flag chart at the end of the paper that is not labeled as Figure 1. I found this chart helpful. The words in the circles that represent government ministries are a bit unclear in graphic. I found it interesting that there are 7/10 ASEAN nations on this chart and the one workshop participant missing is Malaysia who were workshop hosts and survey responders - the participants from Malaysia may not have identified the government engagement??

Water Safety or drowning prevention plans:

Drowning Data:

At line 241 there is a transition from drowning data to challenges to drowning prevention - consider a new subheading at this line; Challenges to Drowning Prevention:

Discussion:

Line 258 - consider the words "has resulted" instead of "necessitates"

Line 261 - consider removing the word "However" and starting with As a multi-sectoral ----

Strengths and Limitations:

Consider a comment on the wide scope or degree of engagement in drowning within the survey and workshop participant group. Some may have been assigned their participation and others have active action for a long time and have participated in the growth of global drowning prevention awareness and activities.

Conclusion: well done

Thank you for this opportunity to review your good work.

6. PLOS authors have the option to publish the peer review history of their article (what does this mean?). If published, this will include your full peer review and any attached files.

Reviewer #1: **Yes: **Pengpeng Ye

Reviewer #2: **Yes: **Stephen B Beerman

---

## [Author Response · Author response to Decision Letter 0]

23 Apr 2024

Thank you for the opportunity to revise the manuscript. We have responded and actioned where needed, each of the reviewer comments in the revised manuscript and the letter of response to reviewers already uploaded.

---

## [Editor Report · Decision Letter 1]

7 May 2024

Drowning Prevention challenges and opportunities: An exploratory study of perspectives of delegates from ASEAN nations

PONE-D-24-10409R1

Dear Dr. Peden,

We’re pleased to inform you that your manuscript has been judged scientifically suitable for publication and will be formally accepted for publication once it meets all outstanding technical requirements.

Kind regards,

Jasmin C Lawes, Ph.D.

Academic Editor

PLOS ONE
---

## [Editor Report · Acceptance letter]

11 May 2024

PONE-D-24-10409R1 

PLOS ONE

Dear Dr. Peden, 

I'm pleased to inform you that your manuscript has been deemed suitable for publication in PLOS ONE. Congratulations! Your manuscript is now being handed over to our production team.

Kind regards, 

on behalf of

Dr. Jasmin C Lawes 

Academic Editor

PLOS ONE